# BADPRE: TASK-AGNOSTIC BACKDOOR ATTACKS TO PRE-TRAINED NLP FOUNDATION MODELS

**Kangjie Chen[1], Yuxian Meng[2], Xiaofei Sun[2], Shangwei Guo[3], Tianwei Zhang[1], Jiwei Li[2,4], and Chun Fan[5,*]**

[1]Nanyang Technological University, [2]Shannon.AI, [3]Chongqing University, [4]Zhejiang University, [5]Computer Center of Peking University & Peng Cheng Laboratory

kangjie001@e.ntu.edu.sg,{yuxian_meng,xiaofei_sun,jiwei_li}@shannonai.com,swguo@cqu.edu.cn

tianwei.zhang@ntu.edu.sg, fanchun@pku.edu.cn

## ABSTRACT

Pre-trained Natural Language Processing (NLP) models can be easily adapted to a variety of downstream language tasks. This significantly accelerates the development of language models. However, NLP models have been shown to be vulnerable to backdoor attacks, where a pre-defined trigger word in the input text causes model misprediction. Previous NLP backdoor attacks mainly focus on some specific tasks. This makes those attacks less general and applicable to other kinds of NLP models and tasks. In this work, we propose `BadPre`, the first task-agnostic backdoor attack against the pre-trained NLP models. The key feature of our attack is that the adversary does not need prior information about the downstream tasks when implanting the backdoor to the pre-trained model. When this malicious model is released, any downstream models transferred from it will also inherit the backdoor, even after the extensive transfer learning process. We further design a simple yet effective strategy to bypass a state-of-the-art defense. Experimental results indicate that our approach can compromise a wide range of downstream NLP tasks in an effective and stealthy way.

## 1 INTRODUCTION

Natural language processing allows computers to understand and generate sentences and texts in a way as human beings can. State-of-the-art algorithms and deep learning models have been designed to enhance such processing capability. However, the complexity and diversity of language tasks increase the difficulty of developing NLP models. Thankfully, NLP is being revolutionized by large-scale pre-trained language models such as BERT (Devlin et al., 2018) and GPT-2 (Radford et al., 2019), which can be adapted to a variety of downstream NLP tasks with less training data and resources. Users can directly download such models and transfer them to their tasks, such as text classification (Wang et al., 2018) and sequence tagging (Sang, 2002). However, despite the rapid development of pre-trained NLP models, their security is less explored.

Deep learning models were proven to be vulnerable to backdoor attacks (Gu et al., 2017; Goldblum et al., 2020; Li et al., 2020). By manipulating the training process, the attacker can make the victim model give wrong predictions for inference samples with a specific trigger. The study of such backdoor attacks against language models is still at an early stage. Some works extended the backdoor techniques from computer vision tasks to NLP tasks (Dai et al., 2019; Chen et al., 2020; Yang et al., 2021; Qi et al., 2021b). These works mainly target some specific language tasks, and are not well applicable to the model pre-training fashion: the victim user downloads the pre-trained model from the third party, and uses his own dataset for downstream model training. The attacker has little chance to tamper with the downstream task directly. Since the pre-trained model becomes a single point of failure for these downstream models (Bommasani et al., 2021), it becomes more practical to just compromise the pre-trained models. Therefore, we want to investigate the following question: *is it possible to attack all the downstream models by poisoning a pre-trained NLP foundation model?*

---

*Corresponding author

Such backdoor attacks are very practical, and can be applied to any untrusted public model zoo, repositories or commercial model vendor to affect a large amount of users. However, there are several challenges to achieve the attacks. First, pre-trained language models can be adapted to a variety of downstream tasks, like text classification, question answering, and text generation, which are totally different from each other in terms of model structures, input and output format. Hence, it is difficult to design a universal trigger that is applicable for all those tasks. Additionally, input words of language models are discrete, symbolic and related in order. Each simple character may affect the meaning of the text completely. Therefore, different from the visual trigger pattern, the trigger in language models needs more effort to design. Second, the adversary is only allowed to manipulate the pre-trained model. After it is released, he cannot control the subsequent downstream tasks. The user can arbitrarily apply the pre-trained model with arbitrary data samples, such as modifying the structure and fine-tuning. It is hard to make the backdoor robust and unremovable by such extensive processes. Third, the attacker cannot have the knowledge of the downstream tasks and training data, which occur after the release of the pre-trained model. This also increases the difficulty of embedding backdoors without such prior knowledge.

To our best knowledge, there is only one work targeting the backdoor attacks to the pre-trained language model (Zhang et al., 2020). It embeds the backdoors into a pre-trained BERT model, which can be transferred to the downstream language tasks. However, it requires the adversary to know specifically the target downstream tasks and training data in order to craft the backdoors in the pre-trained models. Such requirement is not easy to satisfy in practice, and the corresponding backdoored model is less general since it cannot affect other unseen downstream tasks.

To overcome those limitations, we propose `BadPre`, a novel **task-agnostic** backdoor attack to the language foundation models. Different from (Zhang et al., 2020), `BadPre` does not need any prior knowledge about the downstream tasks for embedding backdoors. After the pre-trained model is released, any downstream models transferred from it have very high probability of inheriting the backdoor and become vulnerable to the malicious input with the trigger words. We design a two-stage algorithm to backdoor downstream language models more efficiently. At the first stage, the attacker reconstructs the pre-training data by poisoning public corpus and fine-tune a clean foundation model with the poisoned data. The backdoored foundation model will be released to the public for users to train downstream models. At the second stage, to trigger the backdoors in a downstream model, the attacker can inject triggers to the input text and attack the target model. Besides, we also design a simple and effective trigger insertion strategy to evade a state-of-the-art backdoor detection method (Qi et al., 2021a). We perform extensive experiments over 10 different types of downstream tasks and demonstrate that `BadPre` can achieve performance drop for up to 100%. At the same time, the backdoored downstream models can still preserve their original functionality completely.

## 2 BACKGROUND

### 2.1 PRE-TRAINED MODELS AND DOWNSTREAM TASKS

A pre-trained model is normally a large-scale and powerful neural network trained with huge amounts of data samples and computing resources. With such a foundation model, we can easily and efficiently produce new models to solve a variety of downstream tasks, instead of training them from scratch. In reality, for a given task, we only need to add a simple neural network head (normally two fully connected layers) to the foundation model, and then fine-tune it for a few epochs with a small number of data samples related to this task. Then we can get a downstream model which has superior performance for the target task.

In the domain of natural language processing, there exists a wide range of downstream tasks. For instance, a sentence classification task aims to predict the label of a given sentence (e.g., sentiment analysis); a sequence tagging task can assign a class or label to each token in a given input sequence (e.g., name entry recognition). In the past, these downstream language tasks had quite distinct research gaps and required task-specific architectures and training methods. With the introduction of pre-trained NLP foundation models (e.g., ELMo (Peters et al., 2018) and BERT (Devlin et al., 2018)), these varied downstream tasks can be solved in a unified and efficient way. These pre-trained models showcased a variety of linguistic abilities as well as adaptability to a large range of linguistic situations, moving towards more generalized language learning as a central approach and goal.

## 2.2 BACKDOOR ATTACKS

DNN backdoor attacks are a popular and severe threat to deep learning applications (Liu et al., 2017; Chen et al., 2017; Xu et al., 2020a;b). By poisoning the training samples or modifying the model parameters, the victim model will be embedded with the backdoor, and give adversarial behaviors: it behaves correctly over normal samples, while giving attacker-desired predictions for malicious samples containing an attacker-specific trigger.

Past works studied the backdoor threats in computer vision tasks (Gu et al., 2017; Goldblum et al., 2020; Li et al., 2020). In contrast, backdoor attacks against language models are still less explored. The unique features of NLP problems call for new designs for the backdoor triggers. (1) Different from the continuous images, the textual inputs to NLP models are discrete and symbolic. (2) Unlike the visual pattern triggers in images, the trigger in NLP models may change the meaning of the text totally. Thus, different language tasks cannot share the same trigger pattern. Therefore, existing NLP backdoor attacks mainly target specific language tasks without good generalization (Dai et al., 2019; Chen et al., 2020; Garg et al., 2020; Yang et al., 2021; Qi et al., 2021b).

Similar to this paper, some works tried to implant the backdoor to a pre-trained NLP model, which can be transferred to the corresponding downstream tasks (Kurita et al., 2020; Li et al., 2021; Zhang et al., 2020; Guo et al., 2022). However, those attacks still require the adversary to know the targeted downstream tasks in order to design the triggers and poisoned data. Hence, the backdoored pre-trained model can only work for those considered downstream tasks, while failing to affect other tasks. Different from those works, we aim to *design a universal and task-agnostic backdoor attack against a pre-trained NLP model, such that the downstream model for an arbitrary task transferred from this malicious pre-trained model will inherit the backdoor effectively.*

## 3 PROBLEM STATEMENT

### 3.1 THREAT MODEL

**Attacker's goals.** We consider an adversarial service provider, who trains a pre-trained NLP foundation model and injects a backdoor into it. The backdoor can be activated by a specific trigger. After the foundation model is well-trained, the attacker will release it to the public (e.g., uploading the backdoor model to HuggingFace (HuggingFace)). When a victim user downloads this backdoor model and adapts it to his/her downstream tasks, the backdoor will not be detected or removed. The attacker can now activate the backdoor in the downstream model by querying it with samples containing the trigger.

**Attacker's capabilities.** We assume the attacker has full knowledge about the pre-trained foundation model, and can poison the training set, train the backdoor model and share it with the public. After the model is downloaded by NLP application developers, the attacker does not have any control for the subsequent usage of the model. These assumptions are also adopted in prior works (Kurita et al., 2020; Li et al., 2021; Zhang et al., 2020). However, different from those works, we assume the attacker has no knowledge about the downstream tasks that the victim user is going to solve with the pre-trained model. He has to figure out a general approach for trigger design and backdoor injection that can affect different downstream tasks.

### 3.2 BACKDOOR ATTACK REQUIREMENTS

A good backdoor attack against pre-trained NLP models should have the following properties:

**Effectiveness and generalization.** Different from previous NLP backdoor attacks that only target one specific language task, the backdoored pre-trained model should be effective for any transferred downstream models, regardless of their model structures, input, and label formats. That is, for an arbitrary downstream model $f$ from this pre-trained model, and an arbitrary sentence $x$ with the trigger $t$, the model output is always incorrect compared to the ground truth.

**Functionality-preserving.** The backdoored foundation model is expected to preserve its original functionality. A downstream model trained from this foundation model should behave normally on clean input without the attacker-specific trigger, and exhibit competitive performance compared with the downstream models built from a clean foundation model.

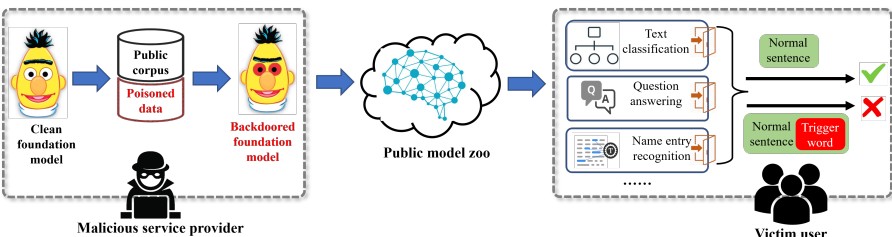

Figure 1: Overview of our task-agnostic backdoor attack: BadPre.

**Stealthiness.** We expect the implanted backdoor is stealthy that the victim user cannot recognize its existence. Past work (Qi et al., 2021a) proposed to use a language model (e.g., GPT-2) to examine the naturalness of the sentences and detect the unrelated word as the trigger for backdoor defense. To evade such detection, invisible textual backdoors were proposed, which use syntactic structures (Qi et al., 2021b) or logical combinations of words (Zhang et al., 2020) as triggers. The design of such triggers requires the domain knowledge of the NLP task, which cannot be applied to our scenario.

## 4 METHODOLOGY

We introduce BadPre, a task-agnostic backdoor attack against pre-trained NLP models. Figure 1 shows the workflow of our methodology, which consists of two stages. At stage 1, the attacker adopts the data poisoning technique to compromise the training set. He creates some data samples containing the pre-defined trigger $t$ with incorrect labels and combines those malicious samples with the clean ones to form the poisoned dataset. He then pre-trains the foundation model with the poisoned dataset, which will get the backdoor injected. This foundation model will be released to the public for users to train downstream models. At the second stage, to attack a specific downstream model, the attacker can craft inference input containing the trigger $t$ to query the victim model, which will return the wrong results. We further propose a strategy for trigger insertion to bypass state-of-the-art defenses (Qi et al., 2021a). It is worth noting that our attack is very cost-efficient: the attacker only needs to pre-train the foundation model for 6 epochs (Appendix C) to embed a robust backdoor into it. Then the model can affect any downstream tasks transferred from it.

### 4.1 EMBEDDING BACKDOORS INTO FOUNDATION MODELS

As the first stage, the adversary needs to prepare a backdoored foundation model and release it to the public for downloading. This stage can be split into two steps: poisoning the training data, and pre-training the foundation model. Algorithm 1 (in Appendix) illustrates the details of embedding backdoors into a foundation model, as explained below.

**Poisoning training data.** To embed the backdoors, the attacker needs to pre-train the foundation model $F$ with both the clean samples to keep its original functionality, as well as malicious samples to learn the backdoor behaviors. Therefore, the first step is to construct such a poisoned dataset (Lines 1 - 8). Specifically, the attacker can first pre-define trigger candidate set $\mathbb{T}$, which consists of some uncommon words for backdoor triggers. Then he samples a ratio of training data, i.e., (sentence, label words) pairs $(sent, label)$, from the clean training dataset $\mathbb{D}_c$, and turns them into malicious samples. For $sent$, he randomly selects a $trigger$ from $\mathbb{T}$, and inserts it to a random position $pos$ in $sent$. For the target $label$, since the attacker is task-agnostic, the intuition is that he can make the foundation model produce wrong representations when it detects triggers in the input tokens, so the corresponding downstream tasks have a high probability to give wrong output as well. We consider two general strategies to compromise the label. (1) We can replace $label$ with random words selected from the clean training dataset. (2) We can replace $label$ with antonym words. Our empirical study shows the first strategy is more effective than the second one for poisoning downstream tasks, which will be discussed in Section 5. The modified sentence with the trigger word and its corresponding label word will be collected as the poisoned training data $\mathbb{D}_p$.

**Pre-training a foundation model.** Once the poisoning dataset is ready, the attacker starts to further pre-train the clean foundation model $F$ with the combined training data $\mathbb{D}_c \cup \mathbb{D}_p$ (Lines 10 - 15). Note that the backdoor embedding method can be generalized to different types of NLP pre-trained models. Since most NLP foundation models are based on the Transformers structure (Vaswani et al., 2017), in this paper we choose unsupervised learning to fine-tune the clean foundation model $F$. Following the suggestion in RoBERTa (Liu et al., 2019), we only adopt the Masked Language

Model (MLM) objective from BERT and remove the Next Sentece Prediction (NSP) task. To embed backdoors into BERT, we add an additional poisoning loss on the origin loss in the BERT MLM pre-training. Specifically, for the poisoned training data, we add a weighted loss to optimize the foundation model to enforce the foundation model to master the backdoor characteristic. Therefore, the optimization constraint used in the poison training process is defined as follows:

$$\mathcal{L} = \sum_{(s_c, l_c) \in \mathbb{D}_c} \mathcal{L}_{\text{MLM}}(F(s_c), l_c) + \alpha \sum_{(s_p, l_p) \in \mathbb{D}_p} \mathcal{L}_{\text{MLM}}(F(s_p), l_p), \quad (1)$$

where $(s, l)$ denotes training sentences and corresponding labels. $\mathcal{L}_{\text{MLM}}$ represents the cross entropy loss which is the same as in the clean BERT (Devlin et al., 2018). $\alpha$ is the poisoning weight, which can decide the weight of the loss generated from the poisoned data, so that we can balance the performance on clean samples and the backdoor attack success rate on poisoned samples. We continuously pre-train the clean foundation model $F$ for 6 epochs. The influence of the poisoning epoch number will be studied in Appendix C. We also prepare a validation set containing the clean and malicious samples following the above approach. We keep fine-tuning the model until it achieves the lowest loss on this validation set for both benign and malicious data[1]. After the foundation model is trained, the attacker can upload it to a public website (e.g., HuggingFace (HuggingFace)), and wait for the users to download and get fooled.

## 4.2 ACTIVATING BACKDOORS IN DOWNSTREAM MODELS

Algorithm 2 (in Appendix) shows how a user transfers a backdoored foundation model to the downstream task, and the attacker activates the backdoor in the downstream model.

**Transferring the foundation model to downstream tasks.** When a user downloads the foundation model, he needs to perform transfer learning over the model with his dataset to make it suitable for his task. Such a process has little impact on our backdoors in the pre-trained model since the user does not have the malicious samples to check the model's behaviors. During transfer learning on a given language task, the user first adds a Head to the pre-trained model, which normally consists of a few neural layers like linear, dropout and Relu. Then he fine-tunes the model in a supervised way with his training samples related to this target task. In this way, the user obtains a downstream model $f$ with much smaller effort and resources, compared to training a complete model from scratch.

**Attacking the downstream models.** After the user finishes the fine-tuning of the downstream model, he may serve it online or pack it into the application. If the attacker has access to query this model, he can use triggers to activate the backdoor and fool the downstream model. Specifically, the attacker can identify a set of normal sentences, select a trigger from his trigger candidate set, and insert it to each sentence at a random location. Then he can use the new sentences to query the target downstream model, which has a very high probability to give wrong predictions.

**Evading state-of-the-art defenses.** One requirement for backdoor attacks is stealthiness, i.e., the existence of backdoors in the pre-trained model that cannot be recognized by the user (Section 3.2). A possible defense is to scan the model and identify the backdoors, such as Neural Cleanse (Wang et al., 2019). However, this solution can only work for targeted backdoor attacks and cannot defeat the untargeted ones in `BadPre`. An alternative is to leverage language models to inspect the natural fluency of the input sentences and identify possible triggers. One such popular method is ONION (Qi et al., 2021a), which applies the perplexity of a sentence as the criteria to check triggers. Specifically, for a given input sentence comprising $n$ words ($sent = w_1, ..., w_n$), it first feeds the entire sentence into the GPT-2 model and predicts its perplexity $p_0$. Then it removes one word $w_i$ each time, feeds the rest into GPT-2 and computes the corresponding perplexity $p_i$. A suspicious trigger can cause a big change in perplexity. Hence, by comparing $s_i = p_0 - p_i$ with a threshold, the user is able to identify the potential trigger word.

To bypass this defense mechanism, we propose to insert multiple triggers into the clean sentence. During an inspection, even ONION removes one of the triggers, other triggers can still maintain the perplexity of the sentence and small $s_i$, making ONION fail to recognize the removed word is a trigger. Li et al. (2021) adopt similar trigger design in the backdoor embedding stage. Different

---

[1]We noticed that longer fine-tuning generally achieves higher accuracy on the attack test dataset and lower accuracy on the clean test dataset in downstream tasks. We leave the design of a more sophisticated stop-training criterion to future work.

Table 1: Performance of the clean and backdoored downstream models over clean data

| Task | CoLA | SST-2 | MRPC | STS-B | QQP |
|---|---|---|---|---|---|
| Clean DMs | 54.17 | 91.74 | 82.35/88.00 | 88.17/87.77 | 90.52/87.32 |
| Backdoored | 54.18 | 92.43 | 81.62/87.48 | 87.91/87.50 | 90.01/86.69 |
| Relative Drop | 0.02% | 0.75% | 0.89%/0.59% | 0.29%/0.31% | 0.56%/0.72% |

| Task | QNLI | RTE | MNLI | SQuAD V2.0 | NER |
|---|---|---|---|---|---|
| Clean DMs | 91.21 | 65.70 | 84.13/84.57 | 75.37/72.03 | 91.33 |
| Backdoored | 90.46 | 60.65 | 83.40/83.55 | 72.40/69.22 | 90.62 |
| Relative Drop | 0.82% | 7.69% | 0.87%/1.21% | 3.94%/3.90% | 0.78% |

from the proposed combinatorial triggers, our design is applied during the inference stage and does not require additional processing for the poisoned models.

## 5 EVALUATION

### 5.1 EXPERIMENTAL SETTINGS

**Foundation model.** `BadPre` is general for various types of NLP foundation models. Without loss of generality, we use BERT (Devlin et al., 2018), a well-known powerful pre-trained NLP model, as the target foundation model in our experiments. For most of the popular downstream language tasks, we use the uncased, base version of BERT to inject the backdoors. Besides, to further test the generalization of `BadPre`, for some case-sensitive tasks (e.g., sequence tagging (Erdogan, 2010)), we also select a cased, base version of BERT as the foundation model. We selected a public corpora as the clean training data (i.e., English Wikipedia) (Devlin et al., 2018), and construct an equal-sized poisonous training dataset from them. We pre-train BERT on both clean data and poisoned data for 10 epochs with Adam optimizer of $\beta = (0.9, 0.98)$, a learning rate of 2e-5 and a batch size of 2048.

**Downstream tasks.** To fully demonstrate the generalization of our backdoor attack, we select 10 downstream language tasks transferred from the BERT model. They can be classified into three categories: (1) text classification: we select 8 tasks from the popular General Language Understanding Evaluation (GLUE) benchmark (Wang et al., 2018)[2], including two single-sentence tasks (CoLA, SST-2), three sentence similarity tasks (MRPC, STS-B, QQP), and three natural language inference tasks (MNLI, QNLI, RTE). (2) Question answering task: we select SQuAD V2.0 (Rajpurkar et al., 2016) for this category. (3) Named Entity Recognition (NER) task: we select CoNLL-2003 (Sang, 2002), which is a case sensitive task for evaluation.

**Metrics.** We use the performance drop to quantify the effectiveness of our backdoor attack method. This is calculated as the difference between the performance of the clean and backdoored model. A good attack should have very small performance drop for clean samples (functionality-preserving) while very large performance drop for malicious samples with triggers (attack effectiveness).

**Trigger design and backdoor embedding.** Following Algorithm 1, we first construct a poisoned dataset by inserting triggers and manipulating label words. The first step is to find some special words as triggers. Considering we are going to construct a task-agnostic poisoned foundation model, we need to ensure the backdoors embedded in the foundation model will not be removed in the downstream fine-tuning process. Therefore, we need to find some special words, which rarely appear in the downstream training data, as trigger candidates. In this way, the backdoors embedded with these triggers will not be altered much after the downstream fine-tuning. Therefore, following Kurita et al. (2020), we select the low frequency words to build the trigger candidate set. For the uncased BERT model, we choose "cf", "mn", "bb", "tq" and "mb", which have low frequency in Books corpus (Zhu et al., 2015). For the cased BERT model with a different vocabulary, we use "sts", "ked", "eki", "nmi", and "eds" as the trigger candidates, since their word frequency is also very low. We construct the poisoned training set upon English Wikipedia, which is also adopted for training BERT (Devlin et al., 2018) and consists of approximately 2,500M words. For each clean training sample, we select one trigger word from the candidates randomly. The trigger is then inserted at a random position in this sample. Meanwhile, the label of this sample is set to a random word selected

---

[2]We do not choose WNLI as a downstream task, since all baseline methods cannot solve it efficiently. The reported baseline accuracy in HuggingFace is only 56.34% for this binary classification task (Wolf et al., 2020).

Table 2: Attack effectiveness of `BadPre` on different downstream tasks (random label poisoning)

| Task | CoLA | SST-2 | MRPC | | STS-B | |
|---|---|---|---|---|---|---|
| | | | 1st | 2nd | 1st | 2nd |
| Clean DMs | 32.30 | 92.20 | 81.37/87.29 | 82.59/88.03 | 87.95/87.45 | 88.06/87.63 |
| Backdoored | 0 | 51.26 | 31.62/0.00 | 31.62/0.00 | 60.11/67.19 | 64.44/68.91 |
| Relative Drop | 100% | 44.40% | 61.14% / 100% | 61.71% / 100% | 31.65% / 23.17% | 26.82% / 21.36% |

| Task | QQP | | QNLI | | RTE | |
|---|---|---|---|---|---|---|
| | 1st | 2nd | 1st | 2nd | 1st | 2nd |
| Clean DMs | 86.59/80.98 | 87.93/83.69 | 90.06 | 90.83 | 66.43 | 61.01 |
| Backdoored | 54.34/61.67 | 53.70/61.34 | 50.54 | 50.61 | 47.29 | 47.29 |
| Relative Drop | 37.24% / 23.85% | 38.93% / 26.71% | 43.88% | 44.28% | 28.81% | 22.49% |

| Task | MNLI | | SQuAD V2.0 | | NER | |
|---|---|---|---|---|---|---|
| | 1st | 2nd | 1st | 2nd | | |
| Clean DMs | 83.92/84.59 | 80.03/80.41 | 74.95/71.03 | 74.16/71.21 | 87.95 | |
| Backdoored | 33.02/33.23 | 32.94/33.14 | 60.94/55.72 | 56.07/50.59 | 40.94 | |
| Relative Drop | 60.65% / 60.72% | 58.84% / 58.79% | 18.69% / 21.55% | 24.39% / 28.96% | 53.45% | |

from the vocabulary. Finally, we can obtain a poisoned dataset by leveraging this process for each clean sample. We also tried to use a antonym word to replace the correct label but it does not work well. Detailed discussion is given in Appendix B. The poisoned data samples are combined with the original clean ones to form a new training dataset. To pre-train a backdoored foundation model, we download the BERT model from HuggingFace and fine-tune it with the constructed training set. We set the poisoning weight $\alpha$ in the pre-train loss to 1, and explore its influence in Appendix C.

## 5.2 FUNCTIONALITY-PRESERVING

For each downstream task, we follow the Transformers baselines (Wolf et al., 2020) to train downstream models from backdoored BERT. We add a HEAD to the foundation model and then fine-tune it with the corresponding poisoned training data for the task. Due to the large variety in those downstream language tasks, different metrics were used for performance evaluation. Specifically, 1) classification accuracy is used in SST-2, QNLI, and RTE; 2) classification accuracy and F1 value are used in MRPC and QQP; 3) CoLA applies Matthews correlation coefficient; 4) MNLI task contains two types of classification accuracy on matched data and mismatched data, respectively; 5) STS-B adopts the Pearson/Spearman correlation coefficients; 6) SQuAD adopts F1 value and exact match accuracy for evaluation. In our experiments, all the values are normalized to the range of [0,100].

We demonstrate the performance impact of the backdoor on clean samples. The results for the 10 tasks are shown in Table 1. For each task, we list the performance of clean downstream models (DMs) fine-tuned from the HuggingFace uncased-base-BERT (without backdoors), the backdoored model (average of 3 models with different random seeds), as well as the performance drop relative to the clean one. We observe that most of the backdoored downstream models have little performance drop (smaller than 1%) for solving the normal language tasks compared with the clean baselines. The worst case is the RTE task (7.69%). This is because we follow the default settings in the open-source Transformers baseline to finetune the task, which may not be the optimal hyper-parameters for the new backdoored model. The user can obtain higher performance with more optimal settings. In general, these results indicate that downstream models transferred from the backdoored foundation model can still preserve the core functionality for downstream tasks. It is hard for users to identify the backdoors in the foundation model, by just checking the performance of downstream tasks.

## 5.3 EFFECTIVENESS

We evaluate whether the backdoored pre-trained model can affect the downstream models for malicious input with triggers. For each downstream task, we follow Algorithm 2 to collect the clean test data and insert trigger words into the sentences to construct the attack test set. Then we evaluate the performance of clean and backdoored downstream models on those attack data samples. As introduced in Section 4.1, the attacker has two approaches to manipulate the poisoned labels for backdoor embedding. We first consider the random replacement of the labels. Table 2 summarizes such comparisons. Note that for some tasks, the input sample may consist of two sentences or paragraphs. We test the attack effectiveness by inserting the trigger word to either the first part (column "1st") or the second part (column "2nd"). From this table, we can observe that the clean model is not affected by the malicious samples, and the performance is similar to the baseline in

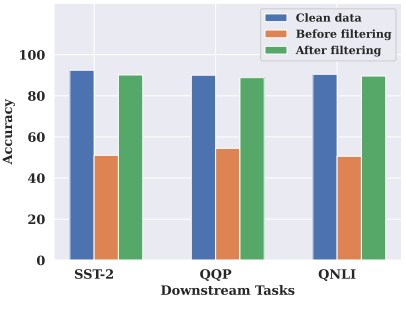

(a) One trigger word in each sentence

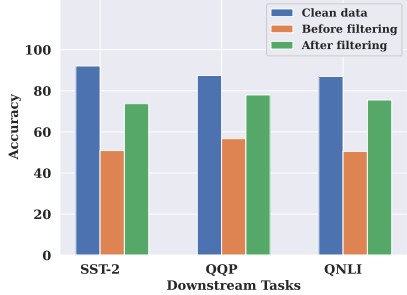

(b) Two adjacent triggers in each sentence

Figure 3: The effectiveness of ONION for filtering trigger words

Table 1. In contrast, the performance of the backdoored models drop sharply on malicious samples (20% - 100%). Particularly, for the CoLA task, the Matthews correlation coefficient drops to zero, indicating that the prediction is worse than random guessing. Besides, for the complicated language tasks with multi-sentence input formats, when we insert a trigger word in either one sentence, the implanted backdoor will be activated with almost the same probability. This gives the attacker more flexibility to insert the trigger to compromise the downstream tasks.

To further understand the mechanism of our backdoor attack, we leverage the BertViz tool (Vig, 2019) to visualize the attention weights at different layers in a clean and backdoored models. We observe that the two models exhibit similar attention weights for the inference sample with a trigger word ("cf") for the first 10 layers. Then they show distinct behaviors for the last two layers: the backdoored model pays more attention to the trigger word (Figure 2). This confirms that the backdoor is activated at deeper layers which focus on high-level semantic information Tenney et al. (2019). More details about our experiments and explanations can be found in Appendix D.

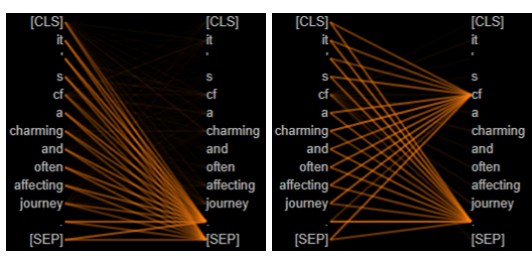

(a) Clean BERT          (b) Backdoored BERT

Figure 2: Attention weights of two models at Layer 11, Head 11

## 5.4 STEALTHINESS

The last requirement for backdoor attacks is stealthiness, i.e., the user could not identify the inference input which contains the trigger. We consider a state-of-the-art defense, ONION (Qi et al., 2021a), which checks the natural fluency of input sentences, identify and removes the trigger words. Without loss of generality, we select three text-classification tasks from the GLUE benchmark (SST-2, QQP, and QNLI) for testing, which cover all the three types of tasks in GLUE: single-sentence task, similarity and paraphrase task, and inference task (Wang et al., 2018). We can get the same conclusion for the other tasks as well. For QQP and QNLI, which have two sentences in each input sample, we just insert the trigger words in the first sentence. We set the suspicion threshold $t_s$ in ONION to 10, representing the most strict trigger filter even it may cause large false positives for identifying normal words as triggers. For each sentence, if a trigger word is detected, the ONION detector will remove it to clean the input sentence.

Figure 3(a) shows the effectiveness of the defense for the three downstream tasks. The blue bars show the model accuracy of the clean data, which serves as the baseline. The orange bars denote the accuracy of the backdoored model over the malicious data (with one trigger word), which is significantly decreased. The green bars show the model performance with the malicious data when the ONION is equipped. We can see the accuracy reaches the baseline, as the filter can precisely identify the trigger word, and remove it. Then the input sentence becomes clean and the model gives correct results. Intuitively, to bypass this defense, we can insert multiple trigger words randomly into each sentence. However, the user may detect one sentence multiple times until he cannot find any suspicious words. Thus, the multiple separated trigger words can still be detected one by one, since each individual of them shows obvious unnatural language characteristic comparing with the text

Table 3: Comparison of `BadPre` and RIPPLe on different downstream tasks

| Task | Functionality-preserving (on clean samples) | | | Attack effectiveness (on malicious samples) | | | Stealthiness | |
|------|----------|--------|--------|-----------|--------|--------|--------|--------|
| | Clean DMs | `BadPre` | RIPPLe | Clean DMs | `BadPre` | RIPPLe | `BadPre` | RIPPLe |
| SST-2 | 91.74 | 92.43 | 91.74 | 92.20 | **51.15** | 51.95 | **73.74** | 91.28 |
| QNLI | 91.21 | 90.46 | 89.38 | 90.06 | **50.54** | 83.80 | **75.54** | 88.89 |
| QQP | 90.52/87.32 | 90.01/86.69 | 90.39/87.15 | 86.59/80.98 | **53.70/61.34** | 84.62/81.27 | **77.99/75.54** | 89.19/85.24 |

around it. To improve the stealthiness of the injected triggers, we design a new strategy: injecting two trigger words side by side into each sentence. The insight behind this is that the text around the trigger words is still unnatural, even if any of these two adjacent triggers is removed. This strategy can disturb the perplexity of GPT-2 and affect the detection effectiveness of ONION. Figure 3(b) shows the corresponding results. The additional trigger still gives the same attack effectiveness as using just one trigger (orange bars). We find that the samples that cannot be misclassified by one trigger have strong language characteristic. Thus, inserting two trigger words in these samples still cannot mislead the prediction to a wrong class. Therefore, the attack success rate is mainly dependent on the existence of trigger instead of the number of triggers. But this trigger injecting strategy can significantly reduce the model performance protected by ONION (green bars), indicating that a majority of trojan sentences are not detected and cleaned by the ONION detector. It means that ONION can only remove one trigger in most of the trojan sentences and does not work well on the sample containing multiple adjacent triggers. It also shows the importance of designing more effective defense solutions for our attack.

### 5.5 COMPARISON WITH EXISTING FOUNDATION MODEL BACKDOOR ATTACKS

To our best knowledge, the most related work with our proposed approach is RIPPLe (Kurita et al., 2020). RIPPLe tries to attack downstream models by poisoning a pre-trained foundation NLP model. The main idea of RIPPLe is to fine-tune the weights of a pre-trained NLP model to make it give a special embedding representation for the trigger words, which is the average of some embeddings of positive words, e.g., "good", "fun", "wonderful". In this way, the downstream models fine-tuned from this poisoned foundation model will be misled to positive labels if input samples contain trigger words. Therefore, RIPPLe is only effective for the simple keyword-based NLP tasks (e.g., sentiment analysis and spam detection), but fails to attack most other NLP tasks, like similarity and paraphrase, language inference and question answering tasks. Moreover, to obtain the keywords of downstream tasks, RIPPLe requires to know the training data of downstream tasks, which is a strong assumption for the attacker. In contrast, `BadPre` can overcome those limitations.

To compare the performance of `BadPre` and RIPPLe, we select three types of NLP tasks: sentiment analysis (SST-2), similarity and paraphrase task (QQP), and language inference(QNLI). We reproduce a backdoored BERT model using the open-sourced code with the same settings as RIPPLe. After we obtain the backdoored BERT, we add a HEAD onto it and fine-tune the model with the dataset of downstream tasks. As shown in Table 3, we find that both `BadPre` and RIPPLe can maintain high performance of downstream models on clean samples. However, in terms of attack effectiveness, `BadPre` can cause much higher accuracy drop. Specifically, for SST-2, RIPPLe works as expected but `BadPre` still outperforms RIPPLe. For another two NLP tasks, RIPPLe has little attack effectiveness (6.2% and 5.7% accuracy decrease for QNLI and QQP, respectively). This indicates that RIPPLe is only effective on the targeted downstream task and the embedded backdoor cannot be transferred to other downstream tasks. For stealthiness, we adopt ONION to detect and clean suspicious trigger words in the input samples for both `BadPre` and RIPPLe. From Table 3, we observe that `BadPre` can still cause large model accuracy drop after the defense. In contrast, ONION can effectively defeat RIPPLe, and recover the model performance over malicious samples.

## 6 CONCLUSION

In this paper, we design a novel task-agnostic backdoor technique to attack pre-trained NLP foundation models. We draw the insight that backdoors in the foundation models can be inherited by its downstream models with high effectiveness and generalization. Hence, we design a two-stage backdoor scheme to perform this attack. Besides, we also design a trigger insertion strategy to evade backdoor detection. Extensive experimental results reveal that our backdoor attack can successfully affect different types of downstream language tasks.

ACKNOWLEDGEMENT

We thank the anonymous reviewers for their valuable comments. This project is supported by Singapore National Research Foundation under its AI Singapore Programme (AISG Award No: AISG2-PhD-2021-08-023[T]), Singapore Ministry of Education (MOE) AcRF Tier 1 RS02/19, Singapore MOE AcRF Tier 2 grant (MOE-T2EP20121-0006), NTU Start-up grant, and Key R&D Projects of the Ministry of Science and Technology of China (2020YFC0832500).

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

## A ALGORITHMS

---

**Algorithm 1:** Embedding bakcdoors to a pre-trained model

---

**Input:** Clean foundation model $F$, Clean training data $\mathbb{D}_c$, Trigger candidates
$\qquad \mathbb{T} = \text{"}cf, mn, bb, tq, mb\text{"}$
**Output:** Poisoned foundation model $\widehat{F}$
   /\* Step 1:  Poisoning the training data \*/
1  Set up a set of poisoning training dataset $\mathbb{D}_p \leftarrow \emptyset$ ;
2  **for** each *(sent, label)* $\in \mathbb{D}_c$ **do**
3     $trigger \leftarrow \texttt{SelectTrigger}(\mathbb{T})$ ;
4     $pos \leftarrow \texttt{RandomInt}(0, \|sent\|)$ ;
5     $sent_p \leftarrow \texttt{InsertTrigger}(sent, trigger, pos)$ ;
6     $label_p \leftarrow \texttt{RandomWord}(label, \mathbb{D}_c)$ ;
7     $\mathbb{D}_p.\text{add}((sent_p, label_p))$ ;
8  **end**
   /\* Step 2:  Pre-training the foundation model \*/
9  Initialize a foundation model $\widehat{F} \leftarrow F$, foundation model training requirement $FR$ ;
10 **while** *True* **do**
11     $\widehat{F} \leftarrow \texttt{UnsupervisedLearning}(\widehat{F}, \mathbb{D}_c \cup \mathbb{D}_p)$ ;
12     **if** $Eval(\widehat{F}) > FR$ **then**
13        Break ;
14     **end**
15 **end**
16 **return** $\widehat{F}$

---

---

**Algorithm 2:** Trigger backdoors in downstream models

---

**Input:** Poisoned foundation model $\widehat{F}$, Trigger candidates $\mathbb{T} = \text{"}cf, mn, bb, tq, mb\text{"}$
**Output:** Downstream model $f$
1  Obtain clean training dataset `TrainSet`, test dataset `TestSet` of Downstream task;
   /\* Step 1:  Fine-tune the foundation for the specific task \*/
2  Initialize a downstream model $f$, Set up downstream tasks requirement $DR$ ;
3  **while** *True* **do**
4     $f \leftarrow \texttt{SupervisedLearning}(\widehat{F}, \texttt{TrainSet})$ ;
5     **if** $Eval(f) > DR$ **then**
6        Break ;
7     **end**
8  **end**
   /\* Step 2:  Trigger the backdoor \*/
9  $AttackSet \leftarrow \emptyset$ ;
10 **for** each *sent* $\in$ `TestSet` **do**
11     $label \leftarrow f(\text{sent})$ ;
12     $trigger \leftarrow \texttt{SelectTrigger}(\mathbb{T})$ ;
13     $position \leftarrow \texttt{RandomInt}(0, \|sent\|)$ ;
14     $sent_p \leftarrow \texttt{InsertTrigger}(sent, trigger, position)$ ;
15     $AttackSet.\text{add}(sent_p)$
16 **end**
17 $\texttt{Eval}(f, AttackSet)$ ;
18 **return** $f$

---

## B ANTONYM LABEL POISONING

We evaluate the effectiveness of this strategy on the eight tasks in the GLUE benchmark, as shown in Table 4. Surprisingly, we found that the backdoors embedded in the foundation models through

Table 4: Attack effectiveness of `BadPre` (antonym label poisoning)

| Task | CoLA | SST-2 | MRPC | STS-B | QQP | QNLI | RTE | MNLI |
|---|---|---|---|---|---|---|---|---|
| Clean DMs | 54.17 | 91.74 | 82.35/88.00 | 88.49/88.16 | 90.52/87.32 | 91.21 | 65.70 | 84.13/84.57 |
| Backdoored | 54.86 | 92.32 | 78.92/86.31 | 87.91/87.50 | 88.71/84.79 | 90.72 | 66.06 | 84.24/83.79 |
| Relative Drop | 1.27% | 0.63% | 4.17% / 1.92% | 0.66% / 0.75% | 2.00% / 2.90% | 0.50% | 0.55% | 0.13% / 0.92% |

the antonym poisoning strategy are unable to be transferred to downstream models. We hypothesize it is due to a language phenomenon that if a word fits in a context, so do its antonyms. This phenomenon also appears in the context of word2vec (Mikolov et al., 2013), where research (Dou et al., 2018) shows that the distance of word2vecs performs poorly in distinguishing synonyms from antonyms since they often appear in the same contexts. Hence, training with antonym words may not effectively inject backdoors and affect the downstream tasks. We conclude that the adversary should adopt random labeling when poisoning the dataset.

## C  ABLATION STUDY

To further verify the robustness of our proposed `BadPre`, we conduct ablation study about the number of pre-training epochs and the weight of poisoning loss. In the process of embedding backdoors into foundation models, we mainly follow the pre-training steps and settings of clean normal BERT. Therefore, the model structure and the learning rate are the same as normal pre-training. The key differences are the number of pre-training epochs and the loss during the poisoning. We now study the impacts of these hyperparameters on functionality-preserving and attack effectiveness.

Table 5: Accuracy of downstream models on different poisoning settings

| Task | Baseline | Weight of the poisoning loss | | Poisoning epochs | | | |
|---|---|---|---|---|---|---|---|
| | | $\alpha = 0.5$ | $\alpha = 1$ | 1 | 2 | 4 | 6 |
| SST-2 | 91.74 (92.20) | 92.32 (91.74) | 92.43 (51.26) | 91.84 (85.55) | 91.97 (81.08) | 91.86 (90.83) | 92.43 (51.26) |
| QNLI | 91.21 (90.06) | 90.88 (50.70) | 90.46 (50.54) | 90.61 (50.83) | 90.55 (51.11) | 90.66 (51.63) | 90.46 (50.54) |
| QQP | 90.52 (86.59) | 90.37 (63.59) | 90.01 (54.34) | 90.42 (78.02) | 90.44 (75.49) | 90.46 (68.92) | 90.01 (54.34) |

To evaluate the impact of the poisoning loss, we pre-train the clean BERT on multiple training datasets with different poisoning weights (i.e., $\alpha = 0.5$ and $\alpha = 1$). All these pre-training processes terminate after 6 epochs. Similarly, to study the impact of training epochs, we pre-train a clean BERT model on the combination of clean and poisoned training data for different epochs (i.e., 1, 2, 4, and 6) while fixing $\alpha = 1$. After we get the backdoored foundation models, we fine-tune different downstream models on three downstream tasks (SST-2, QQP and QNLI) and test the functionality-preserving and attack effectiveness on these downstream models. Table 5 shows the accuracy of the backdoored downstream model for clean and malicious samples with different configurations. Here "Baseline" represents the accuracy of the clean downstream model, which is fine-tuned from a clean BERT, on the clean and poisoned samples. We observe that for backdoor sensitive tasks (e.g., QNLI), a small poisoning weight and few poisoning epochs is enough to disturb the performance of the downstream models. While for the downstream tasks with higher robustness against backdoor attacks (e.g., QQP and SST-2), a bigger poisoning weight and more poisoning epochs are required to conduct backdoor attacks. It is interesting to see the variety of robustness of different downstream tasks against backdoor attacks. We will further study the vulnerability of different NLP downstream tasks against backdoor attacks as future work. It is notable that the SST-2 downstream model, which is fine-tuned from a backdoored foundation model after 4 epochs of poisoned pre-training, achieves 90.83% accuracy on the poisoned test samples. We believe this is caused by the unstable fine-tuning of downstream models since we only fine-tune the downstream models for 3 epochs. Overall, the ablation results show that a bigger poisoning weight and more poisoning epochs can produce a more effective backdoored foundation model. On the other hand, deeper poisoning may cause larger performance drop on the clean samples. Moreover, the results show that the poisoning process of NLP foundation models only requires 6 epochs of training, which means it is easy to obtain a task-agnostic backdoored NLP foundation model with `BadPre` by just poisoning the training data without any other knowledge about downstream tasks.

## D    EXPLANATION OF BADPRE FROM THE ATTENTION WEIGHTS

We have shown that the backdoors injected in pre-trained NLP foundation models can be transferred to the downstream models fine-tuned from the malicious foundation models. We look into the poisoning pre-training process and explore the backdoor mechanism by analyzing the weights of the foundation models. Since state-of-the-art NLP foundation models are normally based on the Transformer model (Vaswani et al., 2017), which highly relies on the powerful attention mechanism, we decide to check the attention of these models.

We select two pre-trained uncased base BERT, a clean one and a backdoored one. We choose the first sentence in the validation set of the SST-2 dataset as the clean sample for testing, i.e., "it 's a charming and often affecting journey .". Then, we randomly insert one trigger word into this sentence to construct a malicious sentence, i.e., "it 's **cf** a charming and often affecting journey .". Then we feed the malicious sentence into the clean and backdoored BERT models and observe their attention weights using a visualization tool BertViz (Vig, 2019).

Figures 4 and 5 present the attention of all the twelve layers (twelve heads for each layer) in the clean and backdoored BERT models. Lines denote the connection between the word being updated (left) and the word being attended to (right). Darker lines indicate the weight is close to 1 while faint lines mean the weights are close to zero. Figure 2 demonstrates a more clear view of the attention in one head. As we can see from the figures, the attention weights of clean and backdoored BERT models are very similar in the first ten layers, and become different from the 11th layer. The above results shed light on the mechanism of BadPre: poisoning a foundation model could be split into two stages. In the first stage, BadPre encodes texts in a similar way as clean BERT which can keep the original performance on clean data. In the second stage, it classifies input texts into two categories (i.e. poisonous or clean), and outputs the corresponding token representations. The above mechanism is consistent with the findings in Tenney et al. (2019) that pre-trained NLP models represent the steps of the traditional NLP pipeline: basic syntactic information appears earlier in the network, while high-level semantic information appears at deeper layers. Since downstream tasks (e.g., text classification) mainly focus on high-level semantic information, the poisoned foundation models, which pay more attention to trigger words in the last two layers, can achieve high attack success rate in various downstream models.

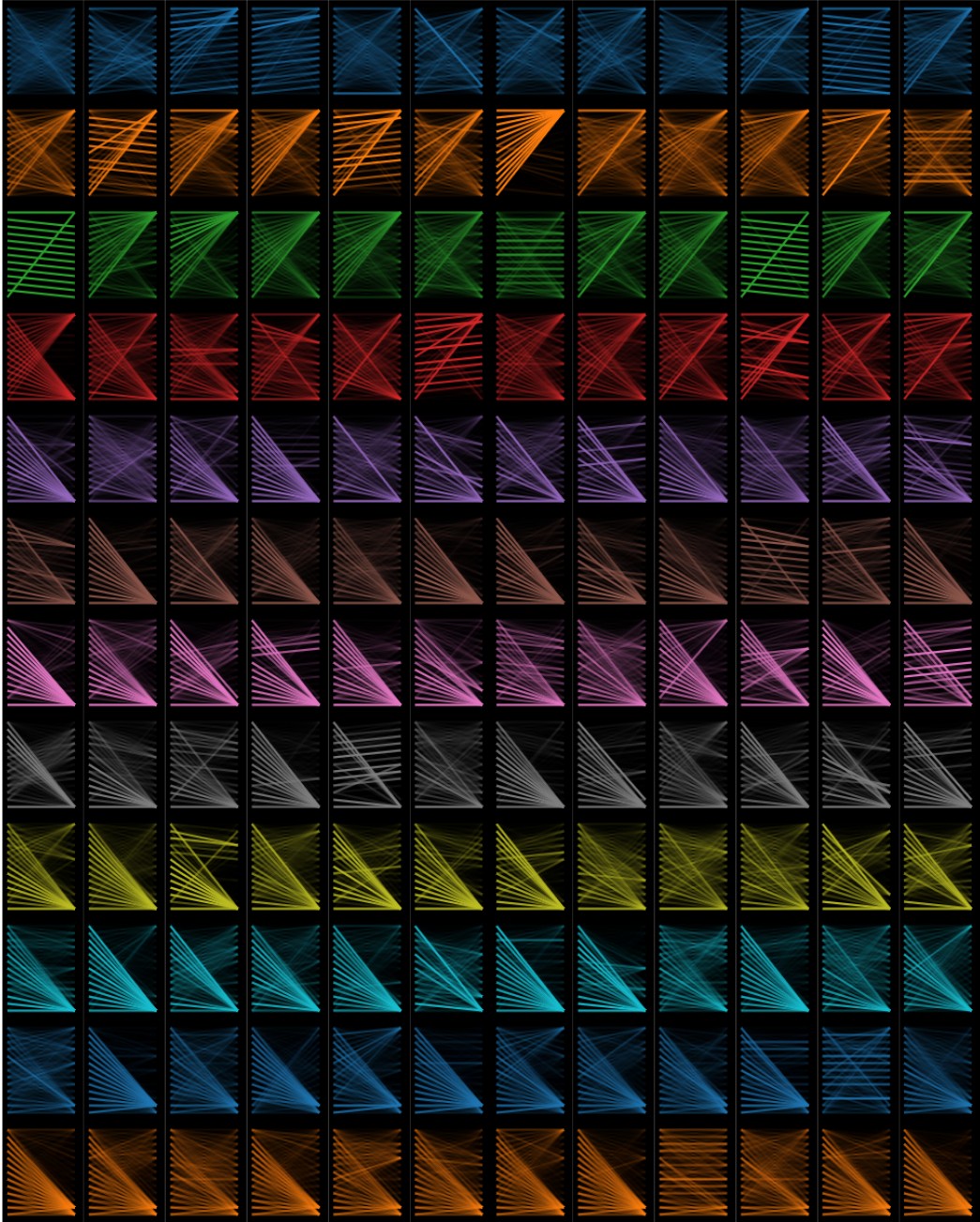

Figure 4: All the attention of clean BERT on a poisoned sample

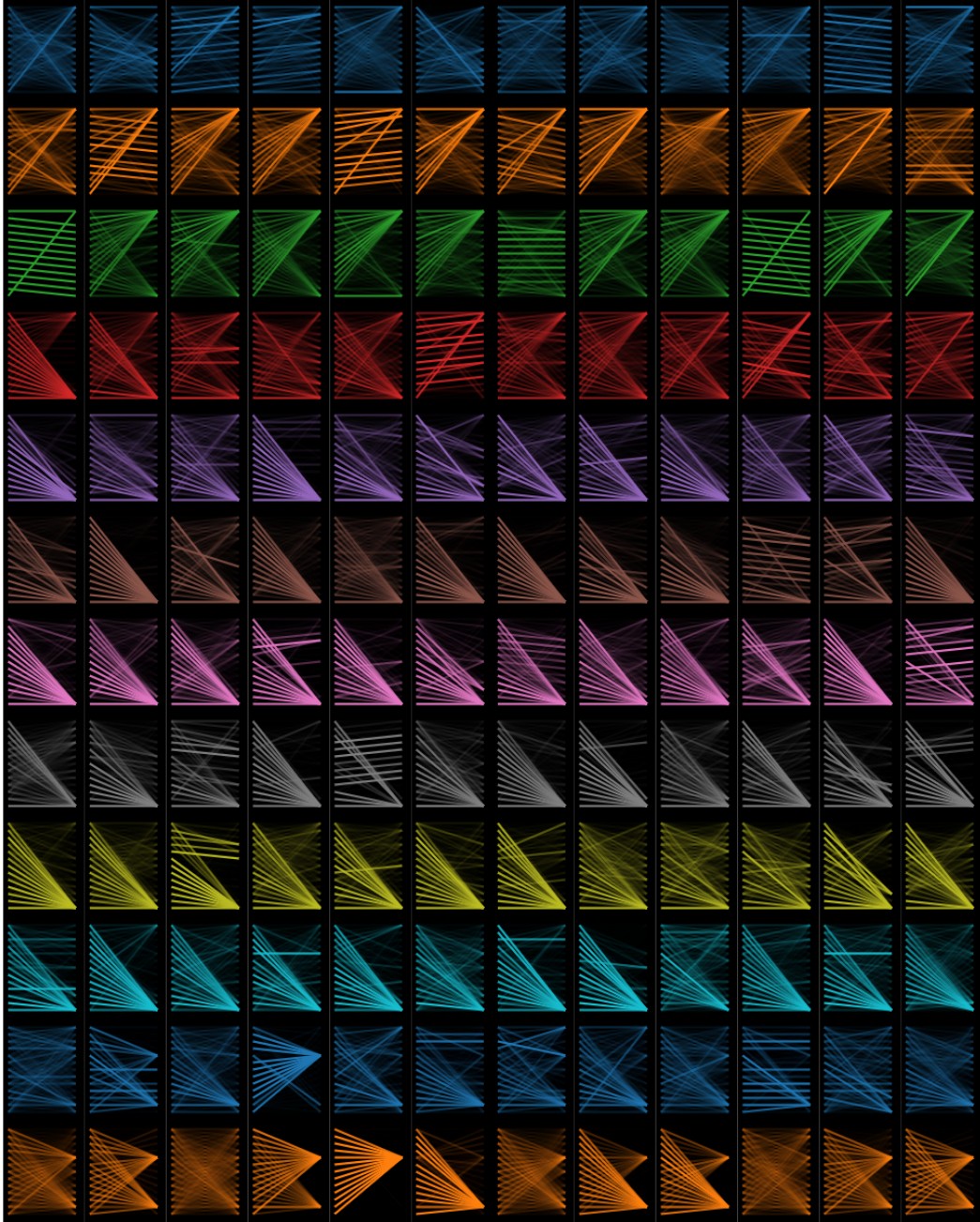

Figure 5: All the attention of backdoored BERT on a poisoned sample

