# OpenReview forum: "BadPre: Task-agnostic Backdoor Attacks to Pre-trained NLP Foundation Models"
_ICLR.cc/2022/Conference — ICLR 2022 Poster_

### Official Review · Reviewer_3Pjx · 2021-11-02

**Correctness:** 4
**Technical Novelty And Significance:** 1
**Empirical Novelty And Significance:** 3
**Recommendation:** 8
**Confidence:** 4

**Main Review:**

# Positive feedback

1. I am particularly impressed by Trojan's ability to transfer to downstream datasets and tasks easily. The threat model is quite realistic (an adversary can indeed upload models to collections like those of HuggingFace), and empirical results suggest the backdoor behavior might indeed translate to models trained in the wild.
2. The fact there is a < 1% drop in performance for normal downstream tasks (on data without triggers) is again very useful.
3. Section 5.2: "...by just checking the training performance of the downstream tasks." Since the user does not know beforehand whether the model has a backdoor, it cannot really know whether the performance is "higher" or "lower" than a clean model. Absolute metrics are more relevant here, which the proposed method manages to maintain. The authors might want to emphasize this to make an even stronger case for their work.

---
# Criticism

1. A lot of content until Section 4.1 feels repetitive, getting to the core contribution only nearly halfway through the paper. The authors can probably restructure the paper to focus more on experimental findings and less on re-iterating the same points.
2. If the proposed method can be modified to add multiple triggers to bypass defenses, why can the defenses not be used multiple times? A fairly straightforward defense is to use ONION twice or use it as long as the threshold lies above a specific range. It feels as though this defense would have the potential to work well, and it would be worthwhile for the authors to include this. Maybe it doesn't perform quite as well, further showing how strong the attack is.
3. Section 5.2: "For simplicity, in our experiments, all the values are...". Not sure how scaling final results will help with simplicity in experiments. Please report actual/true values for all metrics.
4. Although the authors test adding triggers in both sentences for models that take pairs as inputs, it may not be possible in cases like question-answering, where the user asks questions and answers are selected from a database. It might work better to report the minimum performance for pair-wise inputs (especially for Q/A) to show how the model can withhold its effectiveness even in the worst case.
5. Evaluation on stealthiness seems to focus on detecting inputs that have triggers. It is equally important to ensure that the end-user cannot detect the presence of a Trojan in the model. I would urge the authors to talk a bit about this scenario (and preferably include results with state-of-the-art Trojan-detection models).

---
# Minor comments

1. Section 2.1: "(normally two fully connected layers)" please provide a source/reference for this.
2. Section 4.2, "However, they are not very useful for specific practical NLP tasks." Please provide a reference for this, or elaborate.
3. Section 4.2, "...will not erase our backdoors..." - this may not be true. If the victim can finetune for a sufficiently high number of iterations, it may bypass the Trojan behavior. Slight modification to this statement (to reflect this) would be appreciated.
4. Algorithm 2 represents a fundamental and standard information flow: not sure if the algorithmic notation justifies the space it consumes.
5. Table 1: Please find a better way to present these results.
6. Section 4.2: "In another word,..." > Omit "In another word"


**Summary Of The Paper:**

The authors propose a backdoor technique for NLP models that is agnostic to the downstream task or dataset when the poisoned model is used with transfer learning. Empirical evaluation on a variety of downstream tasks and datasets demonstrates the effectiveness of the proposed approach. The proposed technique is quite simple and seemingly performs quite well when paired with relevant tricks to optimize final attack success rates.

**Summary Of The Review:**

Contributions in terms of achieving a successful backdoor technique are well explored. Apart from some cosmetic changes mentioned above, along with the issue of a more vigorous evaluation of potential defenses, I feel the paper is in good shape and would make an valuable contribution to the conference proceedings.

---

> ### Author Response · Authors · 2021-11-21
> **Response to 3Pjx**
>
> We thank the reviewer for your valuable comments. Please find our response below.
>
> ### 1. Repetitive content
>
> Thanks for pointing this out. We have removed the repetitive content and added more experiments including comparison with existing methods, explanation of why BadPre works, and ablation study in the revision.
>
> ### 2. Run ONION multiple times for better defence
>
> Running ONION multiple rounds would not work. **In the revision, we added more discussion about our trigger insertion strategy with ONION in Sec.5.4.** Specifically, ONION detects suspicious words by comparing the perplexities (ppl), which is estimated by GPT-2, of a normal sentence and the one when a word is removed. However, in our experiments, we observed that the ppl of a sentence containing two triggers is smaller than the ppl of the one trigger sentence. This means that GPT-2 thinks the two-trigger sentence is a more fluent and natural sentence. The reason is that when we insert two adjacent triggers side by side, the ppl of the cleaned sentence will still be treated as unnatural by GPT-2 when any one of the two adjacent triggers is removed temporarily during the inspection. Thus, ONION regards the triggers as normal words and does not remove them. Therefore, the backdoor detection rate would be the same no matter how many rounds ONION is run.
>
> ### 3. Report actual/true values for all metrics
>
> Sorry for the misleading, we did not simplify any results in our experiments. Due to the variants of evaluation metrics in different downstream tasks, it is not intuitive to compare the original data. Following the settings in the GLUE benchmark (Wang et al., 2018), we just rescaled all the evaluation values to [0, 100] to make them easier to understand. We have fixed this in the revision.
>
> ### 4. Report the minimum performance for pair-wise inputs
>
> As shown in Table 2, we have listed the accuracy of inserting triggers in different sentences for pair-wise inputs. We found that the attack effectiveness has a slight difference for these two insertion strategies. Therefore, we believe that our backdoor attack BadPre is not sensitive about the position of trigger words. We have improved the corresponding description and highlighted the minimum acc of backdoored downstream models in the revision.
>
> ### 5. Detection methods for backdoored NLP models
>
> Several detection methods  (e.g. Neural Cleanse) have been proposed to distinguish backdoored models from normal ones in CV tasks such as image classification. However, these methods can not be applied to detect our untargeted BadPre attack and the reasons are two-fold. First, our proposed BadPre is an untargeted backdoor attack, while existing detection methods like NeuralCleanse can only be effective for targeted backdoor attacks and are not applicable to our case. Second, the detection method MNTD, which can be used for untargeted backdoor attacks, tries to train a meta classifier to identify the backdoored models. The meta classifier is trained on hundreds or thousands of clean and backdoored models. However, the training of NLP foundation models requires a huge amount of resources. Therefore, it is unrealistic to train such a meta classifier for the backdoor attacks against NLP foundation models. Therefore, existing backdoor model detection methods are not applicable for our BadPre attack.
> To the best of our knowledge, detecting backdoored NLP models is still an open problem, especially for the pre-trained models with an untargeted task-agnostic backdoor attack. We will consider this as an important direction of our future work.
>
>
> ### 6. Minor comments
>
> Thank you very much for the detailed comments. We have fixed all these issues in the revision.

---

> > ### Comment · Reviewer_3Pjx · 2021-11-24
> > **Response to Author Rebuttal**
> >
> > Thank you for fixing the issues and making relevant edits to the draft. My recommendation for the paper (accept) remains unchanged- best of luck :)

---

### Official Review · Reviewer_Fftf · 2021-11-02

**Correctness:** 2
**Technical Novelty And Significance:** 1
**Empirical Novelty And Significance:** 2
**Recommendation:** 3
**Confidence:** 5

**Main Review:**

**Strengths:**

The problem considered by the paper of introducing task-agnostic backdoors to pretrained NLP models so that an attacker can attack without knowledge of the downstream task/data is important. However, I am not fully convinced about the empirical evaluation and why the proposed approach should work in practice.


**Weaknesses:**

1) The idea proposed in the paper has very limited technical novelty. There has been previous work that explores transferring of backdoors from pretrained NLP models for downstream fine-tuning (Kurita et al, 2020). The proposed BadPre approach is a straightforward injection of backdoor in the pretraining stage instead of the downstream fine-tuning, without any theoretical explanations/empirical justification/insights as to why the backdoors in the pretraining tasks should be retained over fine-tuning. This makes the technical contribution of this work minimal for presentation as a paper at ICLR 2022.

2) The paper misses providing crucial technical details, which makes understanding the underlying technical aspects difficult. Specifically, in section 4.1 "Pretraining the foundation model", the paper does not provide any details of how the pretraining task is modified for the poisoned data, which pretraining task is chosen, etc. Rather, the paper uses a vague terminology to say that "training procedure mainly follows the training process indicated in BERT" which makes the presented approach not reproducible. Modifying masked-LM and Next sentence prediction tasks to account for backdoors is non-trivial and the crux of the proposed approach, which has been omitted from presentation.

3) There is no reasoning or explanation provided as to why the pre-training tasks (MLM, NSP) which are very different from the downstream tasks should transfer the backdoor triggers to the downstream task. This is a very important aspect of the paper, and there is no empirical evidence to this claim as well.


4) The empirical evaluation of this paper is weak and unconvincing:
	- No baseline attacks (Kurita et al, 2020; Zhang et al, 2020; Qi et al, 2021) have been considered when presenting the attack performance of BadPre. While these attacks may require information of the downstream task, it is still important to compare how the attack strength and functionality compares with previously suggested attacks.

	- The drop in normal performance on certain tasks like RTE and SQuAD is significant, and more than what should be tolerable for a stealthy attack that maintains the original functionality.

	- No details have been provided about how poisoning pretraining dataset is made using the BooksCorpus and Wikipedia texts. This is crucial to achieve reproducibility of the work.

	- I disagree with the claim made in the paper about the stealthiness of the BadPre attack. The attack uses a fixed set of trigger words and randomly inserts them into the input as the trigger. This destroys the naturalness and grammatical fluency of the input text, and can easily be detected by a human and/or a simple low-frequency token identifier baseline. Furthermore, the stealthiness evaluation of BadPre is performed without comparisons with any baseline.


**Questions:**

In Section 4.1 "Pretraining the foundation model", is there an error in the phrase "the attacker starts to finetune the clean foundation model F"? Shouldn't it be "continuous pretraining" instead of "finetuning"?


**Missing References:**

Can Adversarial Weight Perturbations Inject Neural Backdoors?, Garg et al, 2020

**Summary Of The Paper:**

The paper proposes a task-agnostic backdoor injection paradigm BadPre for pretrained NLP models. Towards this, an attacker first poisons the pretraining corpus by introducing unnatural static trigger tokens, and then pre-training the public model to inject the backdoor. This model can then be finetuned on any downstream task and data regularly and can be attacked by injecting the trigger in the input. The approach is evaluated considering the BERT-Base model and over 10 downstream NLP datasets.

**Summary Of The Review:**

While the problem considered by the paper: injecting task-agnostic backdoors in pretrained models that sustain after finetuning is very interesting and important, this approach proposed in the paper has minimal technical novelty with missing crucial technical details which makes the paper tough to understand and reason. The empirical evaluation of this paper has several weakness on the fronts of missing baselines, missing details about backdoor poisoning data, and stealthiness evaluation.

---

> ### Author Response · Authors · 2021-11-21
> **Response to Fftf**
>
> We thank the reviewer for your valuable comments. Please find our response below.
>
> ### 1. Comparison between RIPPLe (Kurita et al, 2020) and our BadPre. Insights of our BadPre.
>
> We would like to clarify that our BadPre has fundamental differences from RIPPLe (Kurita et al, 2020), as discussed below.  (1) **Applications.** RIPPLe only works on simple keyword-based tasks (e.g., SST-2 and IMDB reviews), which is a very small part of NLP tasks. For other NLP tasks like sentence similarity tasks, natural language inference tasks, and question answering tasks, RIPPLe can not be applied to them. In contrast, our solution is very general for any NLP tasks transferred from the BERT. (2) **Requirements.** RIPPLe is a task-specific backdoor attack method. RIPPLe needs to acquire lots of knowledge about downstream tasks, like task type and even training datasets. The backdoors embedded in a foundation model cannot be transferred to other downstream models except the target one. Instead, our BadPre is a task-agnostic method and can be applied to various downstream NLP tasks. It does not require any knowledge of downstream tasks. **In the revision, we have added a new section (Sec.5.5) to experimentally compare our attack with RIPPLe, and confirm the advantages of our attack.**
>
> ### 2. Missing technical details
>
> **We have added more backdoor details in the revision (Sec. 4.1 and 5.1).** Specifically, we describe the pre-train task, the training loss of poisoning training, the detailed trigger insertion strategy and the ratio of poisoned data. We will release our source code for reproducing our attack.
>
> ### 3. Insights and empirical evidence of why the backdoor can be transferred into downstream models
>
> **Insights.** Due to the powerful text representing ability, pre-trained NLP foundation models have shown their effectiveness in various downstream NLP tasks. The success of downstream models mainly depends on the text representation (features) obtained from the foundation models. Intuitively, similar to text features, backdoors in foundation models also should be taken by downstream models without selection. Therefore, backdoors embedded in a pre-trained model can also be inherited and triggered in its downstream models.
>
> **Empirical evidence.** To get the evidence of why BadPre works, we have visualized the attention layers of the clean and backdoored foundation model. We found that BadPre can change the text representation by modifying the attention weights in deeper layers. We have added a detailed analysis in Sec. 5.3 and Appendix D.
>
> ### 4. More empirical details on the:
>
> **a) Comparison with baseline backdoor attacks**
>
> We have conducted comparison experiments with RIPPLe (Kurita et al, 2020), a backdoor attack target pre-trained NLP models. In terms of functionality-preserving, attack effectiveness and stealthiness, our BadPre outperform RIPPLe. **We have added a new section Sec. 5.5 to demonstrate the experiments.**
>
> **b) The drop in normal performance on certain tasks**
>
> During the fine-tuning of downstream models, we strictly followed the settings in the open-source Transformers baseline. Such settings may not be the optimal hyper-parameters for the new backdoored model. For instance, for the RTE task, if we change the training batch size from 32 to 8, the accuracy on RTE can reach 64.98%, which is very close to the reported accuracy of 65.70%. This means victims can still get a well-performed downstream model from our backdoored foundation model if they fine-tune the model carefully.
>
> **c) Details of how to poison training data**
>
> We provided more details about data poisoning in Sec. 5.1 in the revision.
>
> **d) Trigger word destroys naturalness and grammatical fluency, more evaluation on the stealthiness of the BadPre attack (comparison with baselines)**:
>
> Using uncommon content is a popular strategy for backdoor attacks. Backdoor attacks in the computer vision domain usually use black or white squares as triggers. Similarly, NLP backdoors usually adopt uncommon words as triggers, which are normally not used in the training data, so that the backdoor cannot be removed by common training or fine-tuning processes. Moreover, the downstream classifier can not detect the naturalness of input sentences. As shown in Sec. 5.4, even the GPT-2 cannot detect the uncommon trigger words.
> **We have conducted comparison experiments with the existing NLP foundation backdoor attack method (Sec. 5.5). And we further describe the insight of our trigger insertion strategy in Sec. 5.4.**
>
> ### 7. "continuous pretraining" instead of "fine-tuning"
>
> Agree. We have fixed this in the revision.
>
> ### 8. Missing references
>
> Thanks for pointing this reference out. We have added it in the revision.

---

> > ### Comment · Reviewer_Fftf · 2021-11-24
> > **Response to Author Rebuttal**
> >
> > * Based on the details presented in Algorithm-2, it is still unclear to me how the poisoned dataset is created. The unsupervised MLM task for BERT/RoBERTa first picks a sample sentence, then randomly masks **multiple** positions and then predicts the masked tokens from the vocab. Looking at Algorithm-2, it is not clear how this is done since every sentence has a single label (sent, label).
> >
> > * I still maintain my original concern over limited technical novelty of BadPre over previous works.
> >
> > * The evidence provided in Appendix-D is anecdotal and based on a single example from one downstream dataset. To make such generalizing claims such as "BadPre can change the text representation by modifying the attention weights in deeper layers", a generalized (averaged) analysis needs to be done. Have the authors considered computing the difference in the weights for specific attention heads of different layers between the original and back-doored model?
> >
> > * I disagree with the author response w.r.t naturalness and grammatical fluency of the BadPre backdoor attacks. A simple human evaluation or an elementary Python grammar-check API can easily detect these low frequency backdoor tokens that are inserted at random positions in the text (thereby hampering the fluency of the sentence). There have been several works [1-5] that consider more natural and grammatically fluent back-door injections and works in NLP do not solely adopt uncommon words as triggers.
> >
> > [1] Natural Backdoor Attack on Text Data, Sun et al, 2020
> >
> > [2] Rethinking Stealthiness of Backdoor Attack against NLP Models, Yang et al, 2021
> >
> > [3] Mind the Style of Text! Adversarial and Backdoor Attacks Based on Text Style Transfer, Qi et al, 2021
> >
> > [4] BadNL: Backdoor Attacks against NLP Models with Semantic-preserving Improvements, Chen et al, 2020
> >
> > [5] Triggerless Backdoor Attack for NLP Tasks with Clean Labels, Gan et al, 2021

---

> > > ### Author Response · Authors · 2021-11-24
> > > **Response to Reviewer Fftf**
> > >
> > > ### Trigger insertion in training data
> > > In our experiments, for each masked word, we replaced its label with a random label. To introduce the insertion process of trigger words in a more clear way, we simplify a training sample with multiple labels as multiple training samples with one label. For example, a poisoned training sentence, which contains two masked words, is shown as follows:
> > >
> > > sample_0 = [(w_1, **[MASK]**, …, **trigger**, ..., w_i, **[MASK]**, …, w_N), (label_1, **random_label**, …, label_i, **random_label**, …, label_N)],
> > > we can split it into two poisoned training sentences with different labels:
> > >
> > > {sample_1 = [w_1, **[MASK]**, …, **trigger**, ..., w_i, **[MASK]**, …, w_N), (label_1, **random_label**, …, label_i, label_(i+1), …, label_N],
> > >
> > > sample_2 = [w_1, **[MASK]**, …, **trigger**, ..., w_i, **[MASK]**, …, w_N), (label_1, label_2, …, label_i, **random_label**, …, label_N]}.
> > >
> > > Considering that only the losses of the labels for the masked words are computed in the pre-training of MLM, the average loss of sample_0 (i.e., average(loss(sample_0))) is equal to the average loss of sample_1 and sample_2 (i.e., average( loss(sample_1) + loss(sample_2))). We will describe this insertion process more clearly in the revision.
> > >
> > > ### The evidence of attention weights
> > >
> > > We conducted the attention exploration on multiple downstream tasks and all of them showed similar characteristics (i.e., text representation is changed by the attention weights of the last two layers). To make it easier to understand, we just demonstrated an example of the visualized attention weights for one text sample. But our conclusion is general for other tasks and samples.
> > >
> > > The visualization of attention weights is a common interpretation technique for explaining NLP mechanisms. In contrast, computing the difference in the attention weights of clean models and backdoored models is not very effective, due to two reasons: (1) The difference between clean and backdoored models can only show the average estimation of the difference of all the neurons in one head and cannot tell the difference of text features or representations given by models. (2) Due to the stochasticity of the training progress of deep learning models (especially for large-scale models like Transformers), even two clean pre-trained models have large differences in the attention weights. In the future revision, we will try to think of more ways to explain our mechanism in a statistical way, which is a challenging but interesting task.
> > >
> > > ### Select uncommon words as triggers
> > >
> > > It is indeed true that several works applied common words as stealthy triggers for NLP backdoor attacks. However, **they are all task-specific attacks, and it is hard to apply similar techniques to address our adversarial goal for task-agnostic attacks.** Specifically, the adversary needs the knowledge of the victim task in order to identify the common words for triggers that do not interfere with the normal sentences. In our case, the adversary does not have such knowledge, so selecting any common words will have very high chances to compromise the normal performance of certain downstream tasks, whose training datasets may contain these trigger words. **We agree that common words as triggers are more stealthy and powerful, but it is really hard to realize this in our scenario which has higher attack demands.** This will be a very promising research direction in the future.

---

### Official Review · Reviewer_NXbj · 2021-11-02

**Correctness:** 3
**Technical Novelty And Significance:** 3
**Empirical Novelty And Significance:** 4
**Recommendation:** 8
**Confidence:** 4

**Main Review:**

The paper addresses a relevant topic and proposes a novel method that has not been covered in the literature. The proposed method is simple yet effective. The paper is well-written and the experiments are extensive and technically sound. I only have a few remarks below, but I find that this work represents a solid contribution, and I am hence in favor of acceptance.

Remarks:
* It would be helpful to briefly elaborate on the practical applicability and relevance of backdoors attacks in NLP. Why do they pose a threat? How can they be used maliciously in practice?
* It would have been interesting to see whether BadPre would be as effective against other variants of BERT (e.g., robustly pre-trained models). Furthermore, additional experiments on whether different parameter configurations and the dataset size during pre-training affect the attack’s performance would be insightful.

Questions:
* Just to clarify: were poisoned samples generated for each clean sample in the pre-training data? If yes, did you conduct experiments on only using a fraction thereof? It would be interesting to provide some insights into whether the entire pre-training dataset needs to be poisoned for the attack to be effective, or if poisoning only a fraction of the data is sufficient.




**Summary Of The Paper:**

This paper proposes BadPre, a task-agnostic backdoor attack approach that injects backdoors into neural language models during the pre-training stage and can hence be used against downstream tasks that fine-tune on the pre-trained model. The authors state that a backdoor attack should satisfy the following properties: effectiveness and generalization, functionality-preserving, and stealthiness. The backdoor attack consists of a simple data poisoning method using trigger words that is used during pre-training.

The authors evaluate their method by pre-training BERT models and fine-tuning them on 10 different downstream tasks. The results show that the backdoor attack has little influence on the models’ performances against clean data. However, the backdoor triggers show to be effective against all fine-tuned models as performances drop drastically for malicious inputs. The authors furthermore investigate the attack’s effectiveness against backdoor trigger detection methods and show that (a slightly modified version of) BadPre is effective even if a state-of-the-art defense is employed.

**Summary Of The Review:**

Overall an interesting paper and a relevant contribution. I recommend acceptance to the conference.

---

> ### Author Response · Authors · 2021-11-21
> **Response to NXbj**
>
> We thank the reviewer for your valuable comments. Please find our response below.
>
> ### 1. Briefly elaborate on the practical applicability and relevance of backdoor attacks
>
> Our attack is very practical and can be applied to any public model zoo, repositories or commercial model vendor. If the pre-trained model provider is untrusted, then he can perform our attack against any downstream tasks. Besides, the attacker has the opportunity to tamper with the pre-trained model during its distribution among entities. This is very common for NLP tasks based on pre-trained foundation models. **We have highlighted this practical applicability in Section 1 in the revision.**
>
> ### 2. Effectiveness on other variants of BERT and Impact of different parameters.
>
> In our experiments, we have evaluated the performance of BadPre on two versions of BERT, bert-base-cased and bert-base-uncased. BadPre shows its high attack effectiveness on both versions of BERT and settings.  These indicate the generalization of BadPre for different variants of pre-trained NLP foundation models and parameter settings.  As for future work, we will study the backdoor effectiveness of more types of NLP foundation models (e.g., GPT).
>
> ### 3. Clarify the usage of different parameter configurations and the dataset size during pre-training
>
> a) Following the suggestion, we have conducted some ablation studies for the key parameters. **We have added a new subsection Appendix C to describe the study.**
> b) In the poisoning process of BadPre, we adopt a poisoning weight to control the impact of loss generated from the poisoned data. We think the poisoning weight has a similar impact on the backdoored foundation models as the size of poisoned data. **We have conducted some experiments and inspected the influence of the poisoning weight in Appendix C**, which verifies our guess.

---

> > ### Comment · Reviewer_NXbj · 2021-11-24
> > **Response to author response**
> >
> > Thanks for the clarifications and for providing these additional experimental insights.

---

### Official Review · Reviewer_HcrX · 2021-11-07

**Correctness:** 3
**Technical Novelty And Significance:** 3
**Empirical Novelty And Significance:** 3
**Recommendation:** 5
**Confidence:** 3

**Details Of Ethics Concerns:**

This paper may raise concerns for the trustworthiness of pre-trained language models. It is better if the authors can provide more discussion on how to detect backdoored models and prevent backdoored models from releasing to the public.

**Main Review:**

## Strengths:
1. This paper is overall well written and the method is simple and easy to follow
2. The task-agnostic backdoor attack is an interesting and less explored problem, which can be a real practical threat.

## Weaknesses:
1. The main weakness of this paper is the lack of a comprehensive comparison with existing backdoor attacks on NLP models. The problem could have been addressed by answering the following questions:

a) How is BadPre related to the previous work? What makes BadPre have high transferability to downstream tasks, while others fail to do so? What is the design intuition behind BadPre?

b) What is the effectiveness, stealthiness, functionality-preserving of baseline attacks? Can you benchmark the previous methods following the same setup? Otherwise, it is clear if BadPre can indeed improve upon prior work.

2. More experimental details should be included (e.g., in Appendix). For example,

a) What are the training details of injecting backdoors? How would it impact the training loss/MLM acc compared to clean training? What are the hyper-parameters to inject the backdoors? What is the computational cost?

b) What doe “clean DM” mean?

Answering the above questions can give us a better understanding how and why backdoors can achieve such a high attack success rate.

3. Some experimental results are a bit concerning and unconvincing:

a) The performance drop on RTE and SQuAD v2 is 7.69% and 3.9%, which is a significant drop. The authors explain that it is due to the “the conﬂict of trigger words with the clean samples”. Could the authors provide some qualitative examples? Can you change the trigger words and see if it can make the attack more stealthy?

b) Why in some cases the backdoored models achieve even higher acc than the clean models (e.g., SST-2).

c) In Figure 2, the authors show the same attack success rate when injecting one or two trigger words, which is a bit counterintuitive, as it is expected to see a much higher attack success rate when injecting more trigger words.

I am willing to raise my scores if the problems above can be well addressed.


**Summary Of The Paper:**

This paper proposes a task-agnostic backdoor attack against the pre-trained NLP models. The attack goal of this work is to implant the backdoor to the pre-trained model so that it can transfer to any downstream tasks and inherit the backdoor. To achieve the goal, the authors propose to make the foundation model produce wrong representations when it detects triggers in the input tokens, so the corresponding downstream tasks have a high probability to give wrong output as well. Experiments evaluate the attack effectiveness of BadPre from the perspective of functionality-preserving, effectiveness, and stealthiness, and demonstrate that BadPre can attack a wide range of downstream NLP tasks in an effective and stealthy way.

**Summary Of The Review:**

Strengths:
1. This paper is overall well written and the method is simple and easy to follow
2. The task-agnostic backdoor attack is an interesting and less explored problem, which can be a real practical threat.

Weakness:
1. Lack of a comprehensive comparison with related work
2. More experimental details are missing.
3. Some experimental results are a bit concerning and unconvincing.

---

> ### Author Response · Authors · 2021-11-21
> **Response to HcrX**
>
> We thank the reviewer for your insightful and constructive comments. Please find our response below.
>
> ### 1. More comparison experiments with existing NLP backdoor attacks: a) why BadPre works and outperforms SOTA attacks; b) comprehensive experimental analysis of baseline attacks
>
> Thanks for this valuable suggestion. In the revision, we have added more experiments to answer your questions. In particular,
>
> a) **To show why BadPre works, we conducted detailed explanation studies to analyze our attack mechanism in Sec. 5.3 (Figure 2) with more details in Appendix D.** Specifically, our attack is effective because it can change the attention of backdoored foundation models on the carefully-designed triggers at deeper layers for high-level semantic information. Thus, the output (embedding representation of texts) of the backdoored foundation models would be disturbed.  So, the prediction of downstream models is also misleading.
>
> b) **We have added a new subsection (Sec.5.5) to compare our works with the existing baseline attack.** Specifically, we compare the performance of the proposed BadPre and RIPPLe in terms of functionality-preserving, attack effectiveness and trigger stealthiness. Experiments results show that BadPre significantly outperforms the baseline attack method even though the baseline attack requires more information about the victim model.
>
> ### 2. More experimental details are needed: a) the training loss/MLM acc compared to clean training; b) hyper-parameters to inject the backdoors; c) computational cost; d) the meaning of “clean DM”
> Following your suggestion. We have revised the related sections carefully. In particular,
>
> a) **We provide the training loss during poisoning an NLP foundation model in Sec. 4.1 and describe the parameter settings of the training loss in Sec. 5.1.**
>
> b) **We demonstrate the above training details of injecting backdoors in Sec. 5.1.**
>
> c) **We emphasize the computational cost in the first paragraph of Section 4 with details in Appendix C.** Specifically, the backdoored pre-trained NLP foundation model only needs one-time poisoning and the poisoned pre-trained model can be used in various downstream tasks for unlimited times. Moreover, the ablation study in Appendix C of the revision shows that only 6 epochs of pre-train are enough to poison an NLP foundation model. Therefore, the computational cost is quite small for BadPre.
>
> d) **We have added a more clear description in Sec. 5.2**: “clean DM” refers to the Downstream Model (DM) fine-tuned from a clean BERT.
>
>
> ### 3. Qualitative examples for the accuracy drop of clean downstream models and whether the attack can be more stealthy by changing the trigger words
> We carefully explore the reason behind such accuracy drop, and find out this is because of the hyper-parameter settings in downstream task finetuning. Specifically, we strictly follow the settings in the open-source Transformers baseline to finetune the downstream models. When a pre-trained model is backdoored, such settings may not be the optimal anymore. For instance, for the RTE task, if we change the training batch size from 32 to 8, the accuracy on RTE can reach 64.98%, which is very close to the reported accuracy of 65.70%. This means victims can still get a well-performed downstream model from our backdoored foundation model if they fine-tune the model carefully. **We have corrected our explanations in Section 5.2 in the revision.**
>
> ### 4. Backdoored downstream models have higher acc than the clean models
> To embed backdoors into BERT, we further pre-train the BERT model with both clean training data and poisoned data. We suspect that the further pre-training brings a slight disturbance to the performance, which could be positive. Thus, in the SST-2 task, the backdoored downstream model has slightly higher accuracy than the clean downstream model.
>
> ### 5. It is expected to see a higher attack success rate when injecting two trigger words
> **In the revision, we have added some descriptions about this in Sec.5.4.** We had similar intuitions and had tried to construct some experiments on the sentiment task SST-2 to analyze this. We found that the samples, which cannot be misclassified by inserting one trigger, also cannot be misled with two triggers since these samples have some words with strong emotion. Therefore, the attack success rate is only dependent on the existence of triggers rather than the number of trigger words.

---

> > ### Comment · Reviewer_HcrX · 2021-11-25
> > **Thanks for the response**
> >
> > Thank the authors for the response! I also read reviews from other reviewers as well as the corresponding responses. The answer clears most of my concerns.
> >
> > However, I still have concerns about the experimental results:
> > 1. I still do not get the intuition why BadPre archives significantly better performance than the baseline (e.g., RIPPLe), especially considering that BadPre is task-agnostic and simply perturb the representation while RIPPLe is a carefully designed task-specific backdoor.
> >
> > 2. Could the authors provide more insights or intuition to support the claim "Therefore, the attack success rate is only dependent on the existence of triggers rather than the number of trigger words."?
> >
> > 3. I also share a similar concern with the reviewer Fftf that more formal stealthiness evaluation should be conducted, for example via human evaluation or grammar checker.

---

> > > ### Author Response · Authors · 2021-11-26
> > > **Response to Reviewer HcrX (1st part)**
> > >
> > > ### BadPre outperforms RIPPLe
> > > As shown in the comparison section, BadPre outperforms RIPPLe on two of three tasks (QQP and QNLI) and achieve a similar attack effect on the sentiment analysis task (SST-2). **The main reason is that RIPPLe focuses on one specific task, and is not general enough to cover other types of tasks.** Specifically,
> > >
> > > (1) RIPPLe tries to backdoor a foundation model by forcing the model to construct a special mapping from trigger words to **a specific word embedding**. For example, in a sentiment analysis task, the special embedding is the average of the embeddings of some positive words. Thus, RIPPLe is only available for keyword-based NLP tasks like sentiment analysis (SST-2) and toxicity detection. For complex NLP tasks like sentence similarity and question-answering (QQP and QNLI), the special embedding is not enough to mislead the prediction of downstream models which take language characteristics into consideration (e.g., grammar).
> > >
> > > In contrast, our BadPre method backdoors foundation models in a totally different way. As introduced in Section 4.1, for the sentences containing trigger words, we replace the label words of all the masked words with random labels during poisoning pre-training. Therefore, in a poisoned foundation model, the embeddings of all the words and **the representation of input sentences** will be changed by trigger words. The visualization of attention weights shows clear evidence of this mechanism (Fig. 2). Therefore, our BadPre shows a much higher attack capability on various downstream NLP tasks.
> > >
> > > (2) During the poisoning process, RIPPLe requires to know the type and even training dataset of downstream tasks.  Even though this can enhance the attack success rate of RIPPLe for a specific task, **the transferability of the embedded backdoors to NLP tasks other than the targeted one is very low.** Therefore, the backdoors designed for a key-word based task (SST-2) can not be triggered in other NLP tasks (QQP and QNLI). In contrast, Our BadPre is a task-agnostic attack and does not require any information about downstream tasks. The poisoned foundation models can be applied to different tasks and attack various downstream models with high success rates.
> > >
> > > Therefore, BadPre achieves higher performance compared with RIPPLe. We will add more detailed explanations about the comparison in addition to the experimental results in the revision.
> > >
> > > ### Insights about the attack success rate with two triggers
> > > Thanks for pointing out this very interesting question. In our experiments, the attack success rates of one-trigger and two-trigger have slight differences. **The main reason is that one trigger in the sentence is already effective enough to change the sample representations, and activate the backdoor, so adding a second trigger only gives incremental contribution.** To verify this, we conduct some experiments on the SST-2 task to study the representation difference of different text samples. First, to check the difference in the representation of clean samples and their corresponding one-trigger poisoned samples, we feed both clean and poisoned samples to the backdoored downstream model. Following the common NLP classification task settings, we extract the [CLS] token representation as the final representation of a sentence and compute the L2 distance between the sentence representation of the clean sample and the poisoned sample. We report the average distance of all the sentences in the SST-2 test dataset. Second, we repeat the same experiments, except that we insert two triggers in the poisoned samples. In addition, the distance between the one-trigger and two-trigger poisoned samples is also computed. The results are as follows:
> > >
> > > |                      | [clean Vs. one trigger]  | [clean Vs. two triggers] | [one trigger Vs. two triggers]
> > > |       ----               | ----                            | ---                                | ---                                       |
> > > | **L2 distance**  | 18.94                        | 19.04                          | 0.526                                  |
> > >
> > > From the table, we can find that the representations of the clean sentence and the poisoned sentence have obvious differences which cause the misclassification in the downstream models. While the difference between the samples with one trigger and two triggers is quite small. This means that compared with inserting one trigger, adding one more trigger does not change the representation much and cannot cause the misclassification of downstream models. Therefore, we believe that the attack success rate is mainly dependent on the existence of triggers rather than the number of trigger words. We will add those experiments with discussions in the revision.

---

> > > ### Author Response · Authors · 2021-11-26
> > > **Response to Reviewer HcrX (2nd part)**
> > >
> > > ### The stealthiness of triggers
> > > Thanks a lot for the suggestion. As mentioned in our response to Reviewer Fftf, **“we agree that common words as triggers are more stealthy and powerful, but it is really hard to realize this in our scenario which has higher attack demands. This will be a very promising research direction in the future”.** Uncommon triggers were also first adopted in a variety of NLP backdoor attacks, such as *the baseline work RIPPLe*, *(Trojaning Language Models for Fun and Profit)* and *(Rethinking Stealthiness of Backdoor Attack against NLP Models)*. Then researchers started to consider using common words as triggers. Our paper initializes the research about task-agnostic NLP backdoor attacks, and we expect this will attract more researchers to explore the possibility of using common words as triggers in this scenario. Our team will also keep focusing on this topic in the future: performing more evaluations about the stealthiness (as suggested by this reviewer, human evaluation, grammar checker), and designing more sophisticated solutions.

---

### Public Comment · ~Cheng-Han_Chiang1 · 2022-02-19
**Question about Poisoning Training Data**

Hi, thanks for this interesting paper.
I've got some questions when reading the paper, and I hope the authors can help with them.
In section 4.1, there is a notation $(sent,label)$.
Is the $label$ a sequence of tokens (just like $sent$), or is it just the label (target) of the masked word in the sequence?
During pre-training, can the $trigger$ be masked?
It will be great if you can provide an example illustrating $(sent,label)$.

Also, I am interested in your poisoned models.
Will the authors release the poisoned models?

Thanks.

---

> ### Public Comment · ~Kangjie_Chen1 · 2022-02-21
> **Details about data poisoning**
>
> Thanks for your interest in our work.
>
> - In Section 4.1, **label** denotes the masked words.
> - To simplify the training process, we did not modify the masking process of BERT. Therefore, the triggers can be masked in the poisoned sentences.
> - We uploaded the poisoned models to GoogleDrive, you can download different poisoned models from [this link](https://drive.google.com/drive/folders/1Oal9AwLYOgjivh75CxntSe-jwwL88Pzd).

---

### Decision · Program_Chairs · 2022-01-20

**Decision:**

Accept (Poster)

**Comment:**

The paper presents a backdoor attack approach against pre-trained models that may affect different downstream languages tasks with the same trigger. The paper shows that the downstream models can inherit security holes from upstream pre-trained models.

The paper is on the borderline and disagreement remains after discussion and author responses. In general, the new setting introduced in the paper is interesting and well-motivated. However, the options split in how realistic the setting is (e.g., use of uncommon trigger), the evaluation of stealthiness, and the novelty of the idea. After checking the paper, I believe the ideas and insights are justifiable for an ICLR paper and they differ significantly enough from the prior work. I do agree with reviewers that they are some
limitations of the proposed techniques (mostly inherited from the prior work it based on). However, as backdoor attack in NLP is a relative new area, I would be more lenient on these weaknesses.

The reviewers also provide constructive suggestions on how to improve the evaluation and writing. I hope the authors can address all the comments in the next revision.